# Microgripper Using Soft Microactuators for Manipulation of Living Cells

**DOI:** 10.3390/mi13050794

**Published:** 2022-05-20

**Authors:** Shunnosuke Kodera, Tomoki Watanabe, Yoshiyuki Yokoyama, Takeshi Hayakawa

**Affiliations:** 1Department of Precision Mechanics, Chuo University, Tokyo 112-8551, Japan; kodera_shunnosuke@mnrobo.mech.chuo-u.ac.jp (S.K.); t15.m15.k9.s.wt@mnrobo.mech.chuo-u.ac.jp (T.W.); 2Toyama Industrial Technology Research and Development Center, Toyama 930-0866, Japan; yokoyama@itc.pref.toyama.jp

**Keywords:** microrobot, microgripper, cell manipulation, micromanipulation, thermoresponsive gel

## Abstract

We present a microgripper actuated by a soft microactuator for manipulating a single living cell. Soft actuators have attracted attention in recent years because their compliance which can adapt to soft targets. In this study, we propose a microgripper actuated by soft thermoresponsive hydrogels. The thermoresponsive gel swells in water when the temperature is low and shrinks when the temperature is high. Therefore, the microgripper can be driven by controlling the temperature of the thermoresponsive gel. The gels are actuated by irradiating with infrared (IR) laser to localize heating. The actuation characteristics of the gripper were theoretically analyzed and we designed a gripper that gripped a ≈10 µm size cell. Additionally, we succeeded in actuating the fabricated microgripper with laser irradiation and gripping a single living cell.

## 1. Introduction

Recently, single cell analysis has attracted attention because of its importance in various fields such as medicine, pharmacy, and biology. Various techniques for the manipulation of single cells are required to realize single cell analyses. For example, the trapping and transportation of single cells are necessary to move target cells to a working area. Additionally, trapping to maintain the position of a cell or rotation to control the orientation of a cell are important manipulations to observe target cells from arbitrary directions. Various methods for single cell manipulations have been reported [1,2,3,4,5,6,7,8,9,10,11]. For example, optical tweezers can manipulate a single cell with high positioning accuracy using the optical pressure of a focused laser [1,2]. However, optical tweezers cannot realize strong manipulation forces because optical force is in the order of piconewtons ($10^{-12}$ N) [3]. Other manipulation methods with stronger forces include electric [4], magnetic [5], acoustic [6], or fluidic [7]. These methods use field forces for cell manipulations, simultaneously realizing the manipulation of a large number of cells. However, it is difficult to only manipulate a single cell and realize complex cell manipulations because the precise designs of external fields with a single cell resolution is difficult in principle. It is difficult to realize single cell manipulations with strong force and high positioning accuracy based on non-contact manipulation methods using external fields. Alternatively, mechanical micromanipulators can perform various cell manipulations with a strong force and high positioning accuracy [8]. A mechanical micromanipulator is capable of performing various cell manipulations such as suction, a sting, or the gripping of cells, by changing an end effector. Thus, mechanical micromanipulators can be used for various purposes requiring a strong force. However, they require operator expertise and the method tends to have low throughput and repeatability.

Considering these circumstances, microgrippers are attracting attention to realize cell manipulations with strong force and high accuracy without operator expertise [12,13,14,15,16,17,18,19,20,21]. Conventional microgrippers commonly use piezoelectric or magnetic materials as actuators for driving the grippers. For example, Lofroth et al. proposed an aluminum microgripper driven by a piezoelectric actuator [13]. The microgripper can realize both gripping and cutting by changing the tip shape of the microgripper. Ichikawa et al. proposed a magnetically driven microgripper [22]. This untethered magnetic microgripper realized the manipulation of living oocytes. However, these microgrippers comprise hard materials such as a metal or a silicon, and are difficult to use to manipulate living cells without damage. Although it is possible to install a force feedback for these microgripper systems, the feedback requires force sensors or visual analysis that can make the system complex and unstable.

Such complexity to manipulate soft objects can be avoided using soft materials [23]. Soft materials have compliance because the materials can change and fit their shape to the target objects. Therefore, the manipulation of soft objects such as living organisms can be realized using soft materials without a complicated feedback system. There are also microgrippers using soft materials [24,25,26,27]. Some are driven using magnetic fields [24,25]. However, it is difficult to move individual microgrippers using magnetic fields. Additionally, some microgrippers used soft actuators such as thermoresponsive gels [26]. These microgrippers are driven by controlling the ambient temperatures of the grippers to gently manipulate target objects. However, changing the ambient temperature requires a long time and can damage living cells.

In this paper, we propose a microgripper for manipulating living cells using a soft actuator driven by local heating via laser irradiation. The proposed gripper does not require a complicated feedback system and it has the potential to grip the living cells of varied stiffnesses and sizes. We theoretically show that the proposed microgripper can reduce the forces applied to target cells with a low Young’s modulus. Additionally, the proposed microgripper was fabricated with a sacrificial layer process and it can be driven in an untethered state. The concept and drive method of the proposed microgripper is described in the following section.

## 2. Concept

The proposed microgripper has soft and rigid structures, as shown in Figure 1a. Rigid structures act as the supporting bodies and parts that grip the cells, and soft structures act as soft actuators. The SU-8 (SU-8, Nippon Kayaku Co., Ltd., Tokyo, Japan) hard photoresist was used for the rigid structures. Thermoresponsive poly(N-isopropylacrylamide) (PNIPAAm) gel was used as the soft actuator. Photoprocessable PNIPAAm (BioResist^®^, Nissan Chemical Corp., Tokyo, Japan) [28] was used in the fabrication of the microgripper. BioResist swells by absorbing water at temperatures less than 32 ${}^{\circ }$C and shrinks by releasing water at temperatures greater than 32 ${}^{\circ }$C, as shown in Figure 1b. This temperature-driven volume change of PNIPAAm can be used as a soft actuator.

Local heating with laser irradiation was used to realize the selective actuation of PNIPAAm on the microgripper. BioResist with graphene was used as a light absorber to actuate this soft structure by near-IR laser irradiation. By using laser irradiation, local heating with single cell resolution (≈10 µm) can be realized, as shown in our previous studies [29,30]. The proposed microgripper can grip and release a single cell by switching the laser on and off, as shown in Figure 1a. Additionally, thermal damage to the target cells caused by changing the ambient temperature can be avoided with local heating. The proposed microgripper can reduce the force and strain on the cells because of the compliance of PNIPAAm. Soft actuators receive a reaction force from a cell when the microgripper grips a cell. The soft structure can then reduce the grip force on the cells. The proposed microgripper can then grip cells with less damage to the cells than conventional hard microgrippers.

## 3. Design of the Microgripper

### 3.1. Theoretical Analysis of the Microgripper

First, we perform a theoretical analysis of gripping forces when the microgripper grips a cell to estimate the force and strain on the cell. The analysis model of the microgripper is shown in Figure 2. We assume that the gel actuators and cells are a simple spring in this model. We then set the Young’s modulus of the gel actuators at the center and rear of the gripper as $E_{b}$. We also set the Young’s modulus of cells as $E_{c}$. We define the cross-sectional areas of the cell, where the gels at the center and rear are $A_{c}$, $A_{1}$, and $A_{2}$, respectively. Other geometrical parameters of the microgripper are shown in Figure 2a. The natural lengths of the cell, center, and rear gel actuators are $L_{c}$, $L_{1}$, and $L_{2}$, respectively. Similarly, we define the deformed lengths of a cell, center and rear gel actuators as $L_{c}^{\prime }$, $L_{1}^{\prime }$, and $L_{2}^{\prime }$, respectively. We then define the distances between a cell and center actuator and between two gel actuators as $D_{c1}$ and $D_{12}$, respectively.

The equation of the relationship between the deformed lengths of cells and gel actuators according to the similarity of triangles is shown in Figure 2b.
(1)$$\begin{array}{cc} L_{c}^{\prime }:L_{1}^{\prime } & =D_{Oc}:\left(D_{Oc}+D_{c1}\right) \end{array}$$
(2)$$\begin{array}{cc} L_{1}^{\prime }:L_{2}^{\prime } & =\left(D_{Oc}+D_{c1}\right):\left(D_{Oc}+D_{c1}+D_{12}\right) \end{array}$$

By eliminating $D_{Oc}$ from these equations, we acquire the following Equation (Equation 3).
(3)$$\begin{array}{c} L_{1}^{\prime }\left(D_{12}+D_{c1}\right)=D_{12}L_{c}^{\prime }+L_{2}^{\prime }D_{c1} \end{array}$$

The applied forces on the cell and gel actuators are then in equilibrium while gripping the cell. We define the forces on a cell where the actuator is at the center and rear as $F_{c}$, $F_{1}$ and $F_{2}$, as shown in Figure 2a. The equations of the equilibrium of forces and moment in the microgripper are: (4)$$\begin{array}{cc} \; & F_{1}+F_{2}+F_{c}=0 \end{array}$$(5)$$\begin{array}{cc} \; & F_{1}D_{c1}+F_{2}\left(D_{12}+D_{c1}\right)=0 \end{array}$$

In this model, we assumed that a gripped cell and gel actuators are linear springs. Therefore, we use the following equations from Hooke’s law.
(6)$$\begin{array}{cc} F_{c} & =-E_{c}\frac{A_{c}}{L_{c}}\left(L_{c}^{\prime }-L_{c}\right) \end{array}$$
(7)$$\begin{array}{cc} F_{1} & =-E_{b}\frac{A_{1}}{L_{1}}\left(L_{1}^{\prime }-L_{1}\right) \end{array}$$
(8)$$\begin{array}{cc} F_{2} & =-E_{b}\frac{A_{2}}{L_{2}}\left(L_{2}^{\prime }-L_{2}\right) \end{array}$$

Finally, we acquired the applied force on a gripped cell from Equations (Equation 3)–(8) by eliminating $F_{1},F_{2},L_{1}^{\prime },L_{2}^{\prime },L_{c}^{\prime }$ as follows:(9)$$\begin{array}{c} F_{c}=\frac{\left[L_{c}-\left(1+\alpha \right)L_{1}+\alpha L_{2}\right]Ec}{\frac{L_{c}}{A_{c}}+\left[\frac{{\left(1+\alpha \right)}^{2}L_{1}}{E_{b}A_{1}}+\frac{\alpha^{2}L_{2}}{E_{b}A_{2}}\right]Ec} \end{array}$$

Equation (Equation 9) shows the relationship between the force applied to the cell $F_{c}$ and Young’s modulus of the cell $E_{c}$. Here, $\alpha =D_{c1}/D_{12}$ is a design parameter of the gripper.

Plots of $F_{c}$ in Equation $F_{c}$ of (Equation 9) with various Young’s moduli of cells are shown in black lines in Figure 3a. We vary α= 0.8, 1.0, and 1.2 as a design parameter. The black dashed line shows a plot of α= 0.8, the black solid line shows a plot of α= 1.0, and the black dotted line shows a plot of α= 1.2. In this analysis, we design a gripper as the distance between the gripper tips becomes 0 µm when the gripper is closed without a cell. The size of a target cell is 20 µm and we assume that the shape of the cell is a sphere. Other parameters for the analysis are summarized in Table 1. The design parameters for the proposed soft gripper, such as $L_{1},L_{2},A_{1},\mathrm{and}\;A_{2}$, are determined by experimentally evaluating the sizes of swollen gel actuators, as shown in Table 1, because the expansion ratio is difficult to theoretically estimate. Additionally, we plot the ideal force feedback and displacement feedback grippers in Figure 3a as a green line and a blue line. For these ideal feedback grippers, the target force value for the force feedback gripper is 0.5 µN, and the target strain value for the displacement feedback gripper is 0.4. These plots compare the conventional microgrippers and proposed soft microgripper.

From Figure 3a, the displacement feedback gripper applies a strong force to the cell, especially hard cells. Alternatively, the soft gripper can apply strong forces to cells with a high Young’s modulus, and a weak force to cells with low Young’s modulus in the same design. In other words, this plot shows that the soft gripper can grip a cell with a modulated force according to the Young’s modulus of a cell without feedback. Furthermore, the applied force for the target cells can be changed by changing the design parameter α.

Additionally, Equation (Equation 9) also shows the relationship between the strain and Young’s modulus of a cell. Here, we defined the strain of cell ε as follows and plotted it in Figure 3b.
(10)$$\begin{array}{c} \varepsilon =\frac{L_{c}-L_{c}^{\prime }}{L_{c}} \end{array}$$

From Figure 3b, the force feedback gripper applied an excessively large displacement to the cell with low Young’s modulus. However, the soft grippers can grip a cell with a low Young’s modulus, imparting a small strain to the cell without feedback. Furthermore, the sizes of living cells can vary according to cell cycles. Applied force for various cell sizes are shown in Figure 3c. Considering that normal cell sizes vary from 5 µm to 30 µm, the proposed gripper can grip target cells with not so strong forces. This results shows that the proposed microgripper can adapt the size variation of cells or size change of target cells according to cell cycles.

From these results of the theoretical analysis, the proposed soft microgripper using the gel actuator is suitable for gripping cells with a variety of Young’s moduli without feedback.

### 3.2. Design of the Microgripper

To design the microgripper, we first chose a light absorption material mixed in the gel. Graphene (793663-5ML, Sigma-Aldrich Japan, Tokyo, Japan) was used as a light absorber because it is used for highly efficient photothermal conversions. The temperature of gel actuators mixed with graphene ink can be increased by irradiating with IR laser because mixed graphene can absorb IR light. By increasing the concentration of graphene ink in gel actuators, the gel actuators can have high photoabsorption and a fast response. However, it is difficult to fabricate small patterns of gel actuators mixed with a high concentration of graphene ink. Therefore, we confirmed the processable minimum patterns of gel actuators mixed with different concentrations of graphene ink. We evaluated the line and space of patterned gel actuators.

The fabricated line and space patterns of the BioResist are shown in Figure 4a–c. Figure 4a,b show 40 µm line patterns mixed with 100 µL/g and 200 µL/g graphene ink, respectively. Additionally, Figure 4c is the 10 µm pattern mixed with 100 µL/g graphene ink. As shown in Figure 4a–c, the pattern of (a) succeeded; however, the patterns of (b) and (c) failed because the edges of the line patterns were not clear and line patterns overlapped. The relationships between the processable minimum patterns of gel actuators and concentrations of graphene ink are shown in Figure 4d. As shown in Figure 4d, the minimum size of the patterns of the gel actuator is a trade-off with the response characteristics. The amount of graphene ink must be changed according to the desired application pattern size.

On the basis of the processable minimum pattern of gel actuators, gel actuators with a width of 30 µm were fabricated with high repeatability, as shown in Figure 4e. This width becomes 72 µm when the actuators are swollen. We also designed the length of the center actuator $L_{1}$ as 45 µm and the length of the rear actuator $L_{2}$ as 78 µm. These lengths become 69 µm and 157 µm in the swollen state and these values were used in the analysis of the previous subsection. Furthermore, the thickness of the gripper is designed as 25 µm, which is suitable for gripping a cell with size of 10–20 µm.

## 4. Experiments

### 4.1. Fabrication Process of the Microgripper

The fabrication process of the proposed microgripper is shown in Figure 5. Normal photolithography processes are used for the fabrication processes of the proposed microgripper. However, the proposed microgrippers have to be driven freely. Thus, the proposed microgripper is fabricated with a sacrificial layer process. The material of the sacrificial layer is dextran (Dextran 10910-25, Nalai Tesque Inc., Kyoto, Japan), which is a polysaccharide that is soluble in water and not soluble in organic solvents. Thus, dextran adhered to the substrate during the patterning of the SU-8 and BioResist; however, dextran was removed by water after patterning. The detailed fabrication processes are as follows and as previously described [29].

**(a)** Spin-coating dextran to a glass substrate;**(b)** Spin-coating BioResist to the glass substrate;**(c)** Patterning BioResist;**(d)** Spin-coating SU-8 to the glass substrate;**(e)** Patterning BioResist;**(f)** Immersing the substrate in water;**(g)** Releasing the fabricated pattern in the water;**(h)** Removing the substrate from the water.

### 4.2. Experimental System

The experimental system configuration to irradiate the laser is shown in Figure 6 [30]. We focused IR laser irradiation (1064 nm, max power: 20 W) on the gel actuators using an objective lens (Olympus, Plan C N 10×/0.25 na). Microscopic images were observed and recorded using a CMOS camera (FLIR, BFS-U3-32S4).

## 5. Results

### 5.1. Driving Characteristics of the Microgripper

First, we confirmed that the proposed microgripper can be driven by irradiation with an IR laser. Microscopic images of the drive are shown in Figure 7a,b. As shown in Figure 7a, the microgripper is closed before irradiating the laser. The microgripper is then opened when irradiating it with the laser, as shown in Figure 7b.

We then confirmed the displacement of the microgripper’s tips at various laser powers. We measured distances between the tips as *d* and used a displacement ratio defined by the following equations to evaluate the displacements of the microgripper’s tips.
(11)$$\begin{array}{cc} D_{max}=\; & D_{open}-D_{close} \end{array}$$
(12)$$\begin{array}{cc} \mathrm{Displacement}\;\mathrm{ratio}:\; & \frac{D_{open}-d}{D_{max}} \end{array}$$

The results of the evaluation are shown in Figure 7c. These results confirm the increasing displacement of the gripper’s tips with increasing laser power. In addition, the gripper was nearly opened when the laser power was greater than 120 mW. Therefore, we used a laser power of 120 mW in the following experiments.

We also confirmed the ON/OFF response of the microgripper. The plot of the ON/OFF response is shown in Figure 7d. In the rising phase, the gripper shrank 80% 0.3 s after the irradiation was started. In addition, the microgripper was completely opened 0.75 s after the irradiation was started. In the falling phase, when the irradiation was ceased, the gripper took 1.5 s for 80% swelling and 4 s for 100%. The result of the continuous ON/OFF response is shown in Figure 7e. We evaluated the displacement of the microgripper’s tips with the repeated ON/OFF switching of the laser. We repeatedly turned the laser on for 0.5 s and off for 0.5 s in this evaluation. The microgripper shrank nearly 100% each time. However, the gripper cannot completely swell when the laser is turned off in this evaluation because the response time of the gripper swelling is approximately 4 s, as shown in Figure 7d.

### 5.2. Demonstration of the Gripping a Living Single Cell

Finally, we confirmed that the microgripper can grip a single cell. In this study, Madin–Darby canine kidney (MDCK) cells were used as the target cells. MCDK is a typical cell line that has characteristics of epithelial cells. Thus, MDCK is thought to have an elasticity similar to other organ cells or cancer cells. Target cells were cultured on cell culture dish (VTC-D100, AsOne, Osaka, Japan) with Dulbecco’s modified Eagle medium (DMEM) with (D6046-500ML, Sigma-Aldrich Japan, Tokyo, Japan) 10% fetal bovine serum (FBS). These cells were treated with trypsin to acquire a cell suspension just before the experiments. Acquired cell suspension was introduced into a glass chamber by using a micropipette.

The result of gripping a MDCK cell is shown in Figure 8. First, we opened the microgripper by irradiating it with the laser, as shown in Figure 8a. During laser irradiation, the BioResist shrank and the microgripper was open. In this study, we released target cells into an experimental chamber and waited until a target cell was between the tips of the microgripper. By turning off the laser when the target cell was between the tips, the microgripper gripped the cell and held the cell for approximately 10 s, as shown in Figure 8b. After that, the gripped cell was released by irradiating the actuator with the laser again. The size of the gripped MDCK cell was 14.7 µm before gripping. The size of the MDCK cell became 11.3 µm when it was gripped. The strain of the MDCK cell was 0.23. Here, the theoretical analysis of the soft gripper predicts that the strain of MDCK was 0.25 when the Young’s modulus of MDCK was 5.7 kPa [32] and the Young’s modulus of the BioResist was 2.8 kPa. The difference between the experimental and theoretical strain values was approximately 8.7%.

## 6. Discussions

In this study, we performed the demonstration of the microgripper in room temperature environment. However, temperature control around a microgripper is important because the temperature affects not only the drive characteristics of the microgripper, but also its cell viability. Therefore, herein we discuss the practical use scenario of the proposed microgripper.

Normally, culture cells are incubated in 37 ${}^{\circ }$C and 5% CO_2_ incubator. This temperature is higher than the transition temperature of PNIPAAm, 32 ${}^{\circ }$C. Thus, if we use the microgripper in the culture environment, gel actuators are normally shrunk and the microgripper is normally opened. However, in the field of biology, cells are often treated in 4 ${}^{\circ }$C culture medium when the cells are treated outside of an incubator. This is because the low temperature suppresses metabolisms of cells and prevents necrosis, i.e., cell death caused by environmental changes such as in temperature or pH. The proposed microgripper is intended to be used outside of an incubator because it needs an optical system for actuation. Thus, it is suitable to use a microgripper in a low temperature environment which allows one to use the microgripper as demonstrated in this study.

## 7. Conclusions

In this study, we proposed a soft microgripper actuated with a thermoresponsive gel. The proposed gripper can be driven by irradiating an IR laser and increasing the gel temperature. In addition, we performed the theoretical analyses of the soft gripper. The analyses showed that the proposed soft gripper can grip a cell with less force than conventional hard microgrippers. We succeeded in gripping the living cell with the proposed microgripper. Future work will include adding a moving function to the microgripper and performing experiments with various types of cells of different sizes and Young’s moduli.

## Figures and Tables

**Figure 1 micromachines-13-00794-f001:**
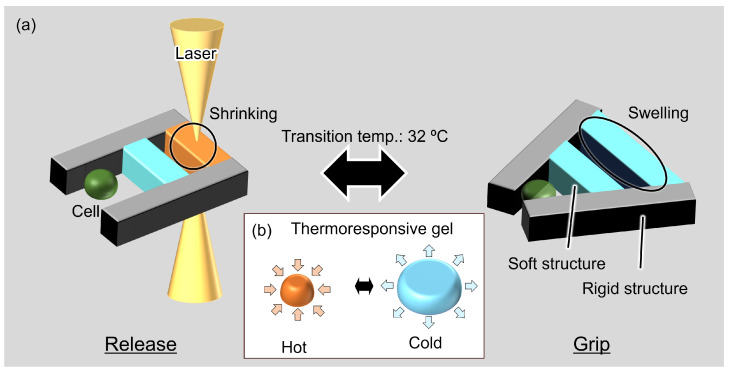
Concept of the proposed microgripper using thermoresponsive gel actuators: (**a**) gripping and driving of the microgripper; and (**b**) the volume change of a thermoresponsive gel as a soft actuator.

**Figure 2 micromachines-13-00794-f002:**
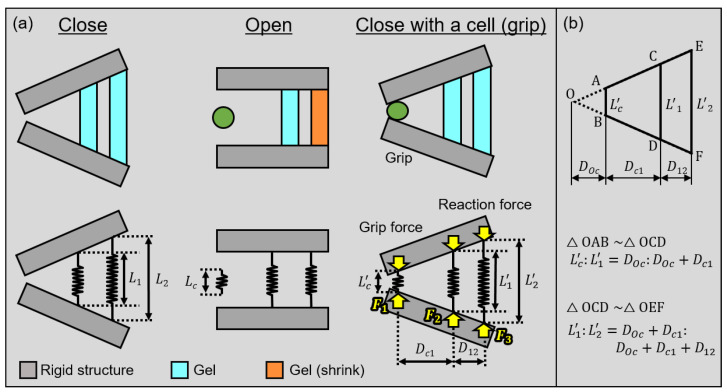
Mechanical model of the microgripper: (**a**) models of each state; and (**b**) similarity relationship.

**Figure 3 micromachines-13-00794-f003:**
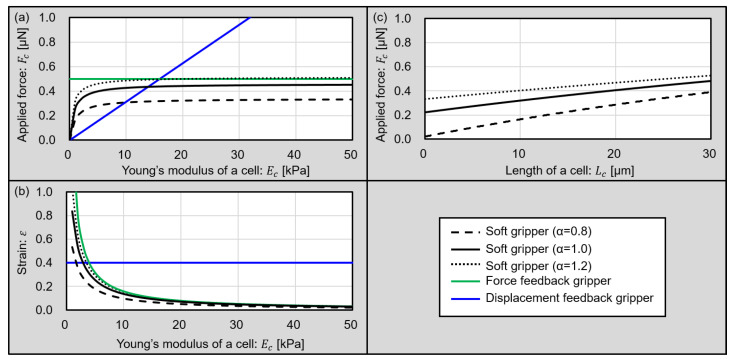
Plot of the theoretical analysis of grippers. (**a**) The relationship between applied forces for target cells and the Young’s modulus of the cells. (**b**) The relationship between the strains of target cells and Young’s modulus of the cells. (**c**) The relationship between applied forces for target cells and the sizes of the cells.

**Figure 4 micromachines-13-00794-f004:**
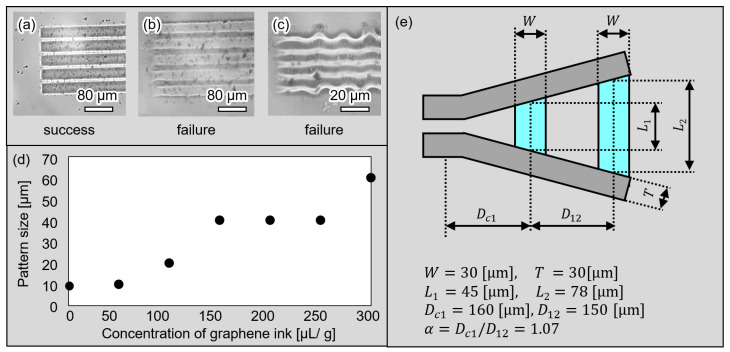
The minimum processable size and design of the microgripper. (**a**–**c**) Examples of the microscopic images of patterned gel actuator; (**d**) minimum processable size with various amounts of graphene ink; and (**e**) the design of the microgripper.

**Figure 5 micromachines-13-00794-f005:**
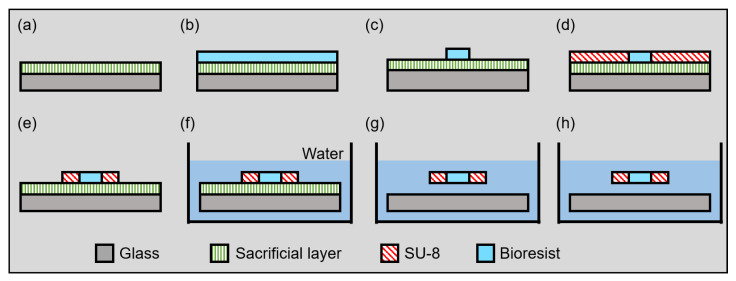
Fabrication process of the microgripper.

**Figure 6 micromachines-13-00794-f006:**
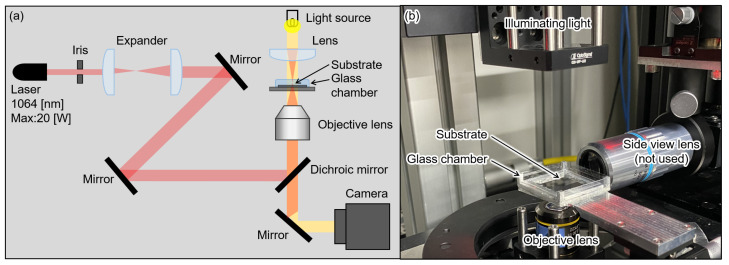
Experimental system. (**a**) System schematic. (**b**) Picture of experimental system.

**Figure 7 micromachines-13-00794-f007:**
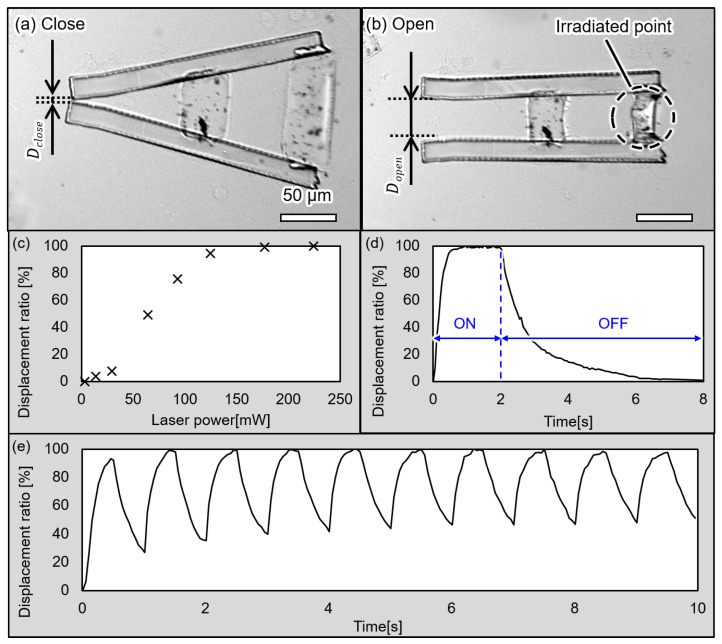
Microscopic images of the microgripper in the (**a**) closed state, (**b**) open state and drive characteristics of the microgripper. (**c**) Displacement ratio at each laser power, (**d**) ON/OFF response of the microgripper, and (**e**) continuous ON/OFF response of the microgripper.

**Figure 8 micromachines-13-00794-f008:**
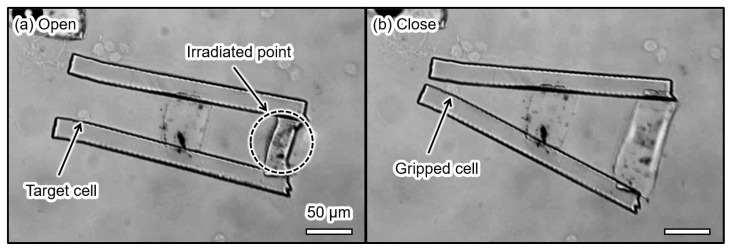
Microscopic images of MDCK gripping by the microgripper (**a**) before gripping and (**b**) during gripping.

**Table 1 micromachines-13-00794-t001:** Parameters for theoretical analysis.

Parameter	Value
$L_{c}$: Length of a cell	20 µm
$L_{1}$: Length of the center gel	69 µm
$L_{2}$: Length of the rear gel	157 µm
$A_{c}$: Cross-section area of a cell	314 µm${}^{2}$
$A_{1}$: Cross-section area of the center gel	1800 µm${}^{2}$
$A_{2}$: Cross-section area of the rear gel	1800 µm${}^{2}$
$E_{b}$: Young’s modulus of a gel	2.83 kPa [31]
α: Ratio of $D_{c1}$ and $D_{12}$	0.8, 1.0, 1.2

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
