# Peer review of "Microgripper Using Soft Microactuators for Manipulation of Living Cells"

_micromachines, 2022, doi:10.3390/mi13050794_

Round 1

Reviewer 1 Report

The manuscript is generally very interesting. However, it needs some revisions before it can be accepted for publication.

Please reposition Figure 1 after row 65 – the first reference for figure 1 is at row 65.

Please reposition Figure 2 after row 89 – the first reference for figure 2 is at row 89.

In equation (1) add parentheses:   Lc’ : L1’ = DOc:(DOc+Dc1)

In equation (2) add parentheses:   L1’ : L2’ = (DOc+Dc1): (DOc+Dc1 +D12):

Please reposition Figure 3 after 100 row – the first reference for figure 3 is at row 100.

Reviewer 2 Report

The authors' work is about a particular kind of microgripper, actuated by "soft". The development of the proposed device is quite complete, it starts from theory and design and ends with fabrication and experimental tests. Results are interesting and worthy of publication in this journal.

Other than checking for minor spelling istakes and typos (for example, "lines" instead of "line s" in line 59), I have a few comments:

Line 105 ("we assume the shape of the cell is a sphere"); lines 115-116 (a sentence is written about different cells with different Young's modulus): The methodology is correct, it's okay to assume a round shape for your theoretical calculations, and to assume that different cells can have different Young's modulus. With that said, a living cell is a lot more than this: it is alive, can change shape and its Young's modulus can also change, fluctuate. I suggest this elements are also addressed in the manuscript.

Section 5.2 "Demonstration of the gripping of a living single cell": can you add more information about MDCK cells, why did you choose them and why are they under study? is there a particular application or reason?

Same section: can you be more clear about the experimental setup? what kind of chamber are you using? how did you release cells?

What is the thickness of the gripper? I think its important to mention it and to correlate it to the size of the target objects

Round 2

Reviewer 2 Report

Thank you for kindly addressing all the suggested elements. I think the manuscript is more complete now. I have only one more comment:

Section 5.2 "Demonstration of the gripping of a living single cell": can you provide some information about the temperature of the cell culture before-during-after the actuation? I think it's crucial to add this, as temperature is important both for the gripper actuation and for the life/health of the cells. I think that, for example, in some scenarios the temperature of the cell culture itself can actuate the gripper without the LASER irradiation, given that the transition temperature is 32°C. Please, kindly address that to improve your manuscript.

Best Regards

Round 3

Reviewer 2 Report

Thank you for kindly addressing all the comments and doubts.

I think the manuscript can be published in its current form